# Oncological Outcomes of Segmentectomy versus Lobectomy in Clinical Stage I Non-Small Cell Lung Cancer up to Two Centimeters: Systematic Review and Meta-Analysis

**DOI:** 10.3390/life13040947

**Published:** 2023-04-04

**Authors:** Ilaria Righi, Sebastiano Maiorca, Cristina Diotti, Gianluca Bonitta, Paolo Mendogni, Davide Tosi, Mario Nosotti, Lorenzo Rosso

**Affiliations:** 1Department of Cardio-Thoracic and Vascular Diseases, Foundation IRCCS Cà Granda Ospedale Maggiore Policlinico, 20122 Milan, Italy; ilaria.righi78@gmail.com (I.R.);; 2Department of Patho-Physiology and Transplantation, University of Milan, 20122 Milan, Italy; 3Department of Thoracic Surgery, European Institute of Oncology, 20141 Milan, Italy

**Keywords:** lung cancer, segmentectomy, lobectomy, lung neoplasms, surgery

## Abstract

Objective. In recent years, pulmonary segmentectomy has emerged as an alternative to lobectomy for the treatment of patients with clinical stage I non-small cell lung cancer. Considering the conflicting results reported in the literature, the oncological effectiveness of segmentectomy remains controversial. To provide new insight into oncological results, we reviewed the literature, including recent randomized trials. Methods. We performed a systematic review for surgical treatment of stage I NSCLC up to 2 cm using MEDLINE and the Cochrane Database from 1990 to December 2022. Primary outcomes for pooled analysis were overall and disease-free survival; secondary outcomes were postoperative complications and 30-day mortality. Results. Eleven studies were considered for the meta-analysis. The pooled analysis included 3074 and 2278 patients who received lobectomy and segmentectomy, respectively. The estimated pooled hazard ratio showed a similar hazard for segmentectomy compared to lobectomy in terms of overall and disease-free survival. The restricted mean survival time difference between the two procedures was statistically and clinically not significant for overall and disease-free survival. Nevertheless, the overall survival hazard ratio was time-dependent: segmentectomy was at a disadvantage starting from 40 months after surgery. Six papers reported 30-day mortality: there were no events on 1766 procedures. The overall relative risk showed that the postoperative complication rate was higher in segmentectomy compared to lobectomy, without statistical significance. Conclusions. Our results suggest that segmentectomy might be a useful alternative to lobectomy for stage I NSCLC up to 2 cm. However, this appears to be time-dependent; in fact, the risk ratio for overall mortality becomes unfavorable for segmentectomy starting at 40 months after surgery. This last observation, together with some still undefined questions (solid/non-solid ratio, depth of the lesion, modest functional savings, etc.), leave room for further investigations on the real oncological effectiveness of segmentectomy.

## 1. Introduction

Immediately after breast cancer (11.7% of the total new cases), lung cancer is the most commonly occurring malignancy worldwide (11.4%). Moreover, lung cancer is the leading cause of death from cancer, accounting for 18.0% of total cancer deaths [1].

Most patients with lung cancer present symptoms only once the disease has become locally advanced or spread to distant organs. Thus, surgery can only be offered to a minority of patients who present with lung cancer in the early stage; actually, only 20.6% of new cases underwent surgery in the United States in 2020 [2]).

The National Institute for Health and Care Excellence guideline, published on March 2019, recommends pulmonary lobectomy for patients with non-small-cell lung cancer (NSCLC) who are well enough and for whom treatment with curative intent is suitable [3]. On the other hand, the diffusion of screening programs increased the rate of diagnosis of small lesions for which the sacrifice of an entire pulmonary lobe may be an excessively aggressive surgical procedure. In addition, alternative therapies such as radiosurgery or thermoablation seem to offer results at least comparable to surgery. To save functioning lung tissue, researchers have begun to verify the oncological efficacy of sublobar lung resections despite the negative results of the historic North American Lung Cancer Study Group study [4]). Pulmonary segmentectomy appears to be oncologically attractive due to the anatomical structure of the segment, which includes a dedicated bronchus and vasculature. Segmentectomy preserves healthy lung tissue, protecting pulmonary function; at the same time, this procedure responds to the principles of an oncologically correct surgery, such as the “en block” resection of the anatomical structure affected by the neoplasm and the respective lymphatic drainage [5]. Conversely, wedge resection leaves lymphatic drainage directed to the lung hilum in place, exposing the patient to local recurrence.

Despite these theoretical premises, controversy remains on the role of segmentectomy in early-stage NSCLC. Some retrospective studies emphasized the disadvantages of sublobar resections, arguing that lobectomy should be considered to be the standard treatment for these tumors. Other studies showed no significant differences in recurrence or survival between these two procedures, suggesting t-segmentectomy as a viable alternative to lobectomy. Several systematic reviews and meta-analyses have indeed been published suggesting that segmentectomy is a valuable alternative treatment for stage I NSCLC up to 2 cm [6,7]. Nonetheless, the results of these reviews were limited by their retrospective nature. Moreover, segmentectomy cannot be considered equivalent for every portion of parenchyma, and often authors do not report which segments and how many lung segments were removed (e.g., very common is the resection of the culmen, which actually includes three segments). Recently, two randomized controlled trials were published [8,9]). These new data need to be analyzed together with the previous ones to shed new light on the controversial topic of the appropriateness of segmentectomy.

The primary objective of the present systematic review and meta-analysis is the assessment of oncological results of pulmonary segmentectomy for stage I NSCLC with a tumor diameter of less than two centimeters. The secondary objective is to verify the safety of the segmentectomy compared to the lobectomy.

## 2. Materials and Methods

### 2.1. Search Strategy and Articles Selection

A systematic literature search was performed in MEDLINE and using the Cochrane Database. The MEDLINE search string was as follows: (((LUNG NEOPLASMS) AND (((segmentectom*[Title/Abstract]) OR (limit* resect*[Title/Abstract])) OR (sublobar[Title/Abstract]))) AND ((intention*[Title/Abstract]) OR (compromis*[Title/Abstract]))) AND ((lung[Title/Abstract]) OR (pulmo*[Title/Abstract])) Filters: English, Adult: 19+ years. The Cochrane Database search terms were the following: “lung cancer” AND segmentect* AND lobectom* in Title, Abstract, and Keyword. The search was limited to a publication date from 1990 until 30 December 2022.

Criteria for considering studies to be included in this review were as follows: 1. Randomized controlled trial, prospective or retrospective study designs; 2. Studies comparing pulmonary lobectomy to segmentectomy; 3. Studies including subjects with NSCLC; 4. Studies reporting overall survival (OS); 5. Outcomes correlated to stage I with tumor with diameter ≤ 2 cm. Exclusion criteria were as follows: 1. Studies including wedge resections; 2. Reviews; 3. Case reports; 4. Letters; 5. Language other than English.

Studies eligibility was independently assessed by three reviewers (IR, SM, and CD), and a fourth reviewer (LR) resolved any disagreements. Figure 1 shows the Preferred Reporting Items for Systematic Reviews and Meta-Analyses (PRISMA) flow chart of the selection process for studies included in the systematic review Preferred Reporting Items for Systematic Reviews and Meta-Analyses (PRISMA) check-list is shown in Appendix A.

### 2.2. Data Extraction

Two reviewers (PM and DT) used a standardized record form to save the extracted data. The following characteristics were collected: first author’s name; year of publication; country; study design; number of patients; gender; age; smoking history; comorbidities; respiratory functional test; ASA score; ground-glass opacity (GGO)—solid tumor rate; postoperative complications and mortality; disease-free survival and overall survival. Two authors (LR and MN) reviewed the database to identify and discuss discrepancies.

### 2.3. Outcomes of Interest and Statistical Analysis

Primary outcomes for pooled analysis were OS and disease-free survival (DFS), and secondary outcomes were the behavior of relative risk for OS and DFS over time, postoperative complications, and 30-day mortality.

Results of the systematic review were summarized qualitatively into a frequentist meta-analysis of pooled hazard ratio (HR) and risk ratio (RR). The inverse-variance random-effects meta-analysis was performed by conventional methods using DerSimonian–Laird estimator to estimate the between-study variance (τ2) [10]. The restricted maximum-likelihood and Q-profile methods were performed in order to estimate τ2. Statistical heterogeneity was evaluated by the Cochran Q-test and I2 index: a value of 25% or smaller was defined as low heterogeneity, between 50% and 75% as moderate heterogeneity, and 75% or larger as high heterogeneity [11]. Wald-type 95% confidence intervals (95% CI) were computed for the pooled measures. The prediction interval for the treatment effect of a new study was calculated according to Borenstein [12]. As the sample size is not the same in all studies, we gradually removed a small sample size to perform a sensitivity analysis to assess the stability of the results (one-leave-out test). Small studies and publication bias effects were assessed by a funnel plot visual inspection and Egger tests for outcomes reported in more than nine studies. The individual patient time-to-event (IPD) data were reconstructed from Kaplan–Maier curves, according to Guyot [13]. Kaplan–Meier curves were digitalized using the Get Data Graph Digitizer software (http://getdata-graphdigitizer.com, accessed on 6 May 2022). HR and relative standard errors were computed by the univariable Cox regression model, and the proportional hazard assumption was checked by means of the diagnostics based on the scaled Schoenfeld residuals. A meta-analysis of restricted mean survival time difference (RMSTD) was performed using a random effect multivariate meta-analysis borrowing strength across time points with a within-trial covariance matrix derived by a bootstrap method with 1000 iterations [14]. Since the sample sizes varied among the studies, we performed the one-leave-out sensitivity test to verify the robustness of the results [15]. Additionally, using IPD, we performed the flexible hazard-based regression model with the inclusion of a normally distributed random intercept. In particular, we modeled the baseline hazard described by the exponential of a B-spline of degree 3 with no interior knots; the model selection was driven according to the Akaike Information Criterion (AIC). The time-dependent effect of surgical treatment was parametrized as interaction terms between the surgical treatment and the baseline hazard and statistically tested by the likelihood ratio test. Hazard function plots were performed using marginal prediction [16]. A Z-score test was performed as appropriate. Two-sided *p*-values were considered statistically significant when less than 0.05, and the CIs were computed at 95%. We used lobectomy as a reference in all statistical analyses. All statistics and graphs were carried out by a professional statistician (GB) using the R software application (version 3.2.2; R Foundation, Vienna, Austria) [17].

### 2.4. Quality and Publication Bias Assessment

Quality assessment of eligible studies was done using the Newcastle–Ottawa Quality Assessment Scale. Each study was judged on eight items categorized into three domains: the selection of the study groups; the comparability of the groups; and the ascertainment of either the exposure or outcome of interest for case-control or cohort studies, respectively. Studies included in this systematic review and meta-analysis reached a score of five or higher.

## 3. Results

Fourteen articles that examined the oncological outcomes of pulmonary segmentectomy versus lobectomy for stage I NSCLC up to 2 cm were identified [7,8,17,18,19,20,21,22,23,24,25,26,27,28] (Table 1). Three papers were excluded [27,28,29] because other articles of the same cohort studies with more cases were already selected [22,25]. Out of the eleven papers included, two were randomized controlled trials, one was a prospective trial with propensity score match analysis, four were retrospective studies with propensity score matching, and four were retrospective studies. Seven articles were written by Eastern authors and included Oriental patients; one article was published by Korean authors, including patients treated in the United States, and finally, three articles were written by Western authors (Table 1). Table 2 presents the Newcastle–Ottawa Quality Assessment Scale for the observational studies included in this meta-analysis.

In this review, a cumulative total of 5352 patients required pulmonary resections for stage I NSCLC up to 2 cm; 2278 patients were treated with pulmonary segmentectomy, whereas 3074 received a lobectomy. The Shanghai Chest Hospital authors stated a mix of tumor location, severe comorbidity, and limited pulmonary function as an indication for segmentectomy [19]. Two studies did not define the selection criteria, but examination of their group characteristics showed a tendency to lower respiratory function in patients with segmentectomy [20,21]. Darras and collaborators offered a segmentectomy if the tumor was located peripherally in a specific segment and the margin was at least two centimeters; the pulmonary function was well-balanced between the two groups despite the retrospective design of the study [24]. Finally, Tane did not state the selection criteria but included lung function as a parameter in his propensity score analysis [26].

### 3.1. Meta-Analysis: Primary Outcomes

The estimated pooled HR calculated with the random effect inverse variance method showed a similar hazard for segmentectomy compared to lobectomy in terms of OS (HR = 0.99; 95% CI from 0.76 to 1.28 *p* < 0.92) on 11 studies, including 5352 patients. Figure 2 shows the forest plot for OS. Heterogeneity was moderate (I^2^ = 38.0%, CI from 0.0% to 69.6%, *p* < 0.09). The funnel plot and the Egger test (*p* = 0.44) did not show evidence of publication bias (Figure 3). Sensitivity analysis showed the robustness of results in terms of point estimation and confidence intervals, whereas the exclusion of Saji’s trial shrinks the heterogeneity to zero without affecting the robustness of results. Visual inspection of Schoenfeld residuals and related global Schoenfeld test for HR calculation did show evidence of proportional hazard assumption violation in four studies, at least.

Evaluation of the restricted mean survival time difference based on Kaplan-Meier curves for OS was possible for all the selected studies. Figure 4 shows the graphic of RMSTD analysis with time horizons. RMSTDs were not statistically nor clinically significant, as detailed in Table 3.

The flexible hazard-based regression analysis showed the non-proportionality between segmentectomy and lobectomy hazard for any cause of mortality (*p* < 0.001). A total of 95% confidence intervals of the two hazards overlapped during the entire time frame considered; therefore, the two surgical treatments had similar hazards (*p* = 0.303) (Figure 5A). On the other hand, the hazard ratio was time-dependent: up to the first 40 months after surgery, the hazard ratio was not significant, but it became significant, assuming a value of about 1.2 starting from month 40 after surgery (Figure 5B).

The estimated pooled HR calculated with random effect inverse variance method showed a similar hazard for segmentectomy compared to lobectomy in terms of DFS (HR = 1.00; 95% CI from 0.78 to 1.27; *p* = 0.97) on 8 studies including 3428 patients. Heterogeneity was low (I^2^ = 18%, 95% CI from 0.0% to 60.5%, *p* = 0.291). The forest plot for DFS is shown in Figure 6. Sensitivity analysis showed the robustness of results when point estimation and confidence intervals were calculated. Visual inspection of Schoenfeld residuals and related global Schoenfeld test for HR calculation did show evidence of proportional hazard assumption violation in three studies at least.

Evaluation of the restricted mean survival time difference based on Kaplan–Meier curves for DFS was possible for eight of the selected studies. Results of the RMSTD analysis with the time horizons are presented in a graphic (Figure 7). RMSTDs were not statistically nor clinically significant, as detailed in Table 4. The flexible hazard-based regression analysis showed the proportionality between the segmentectomy and lobectomy hazard for recurrences (*p* = 0.845). 95% confidence intervals of the two hazards overlapped during the entire time frame considered; therefore, the two surgical treatments had similar hazards (*p* = 0.909) (Figure 8A). Finally, the hazard ratio was time-dependent but not statistically significant when the entire timeframe was considered (Figure 8B).

### 3.2. Meta-Analysis: Secondary Outcomes

Six papers reported 30-day mortality: there were no events in 1766 procedures [8,9,18,20,24,26]. Five articles (2173 patients) reported the number of patients with postoperative complications. The overall relative risk showed that segmentectomy had a postoperative complication rate higher than that of lobectomy, but differences were not statistically significant (RR = 1.14; 95% CI from 0.95 to 1.36; *p* = 0.150) (Figure 9) [8,9,19,20,26]. Despite the fact that polled data were homogeneous (I^2^ = 0%; 95% CI from 0% to 79%; *p* = 0.63), omitting the Saji’s trial by the leave-one-out analysis led to the relative risk for postoperative complications to reach statistical significance (RR = 1.24; 94% CI from 1.02 to 1.52; *p* = 0.029).

## 4. Discussion

Over the past two decades, a growing number of retrospective studies analyzed the possible oncological equivalence of segmentectomy and lobectomy for the treatment of early NSCLC. Segmentectomy, compared to lobectomy, was suggested to achieve comparable oncological results and to better preserve respiratory function. Nevertheless, caution should be exercised when selecting patients for segmentectomy: the size of the tumor is especially important, as more than one study showed unfavorable oncological outcomes in patients treated with segmentectomy for stage I NSCLC greater than two centimeters [20,30]. For this reason, we included in our review only articles related to patients with NSLC up to 2 cm.

The scientific community has long awaited the conclusion of randomized trials that could add scientifically sound information to the choice between segmentectomy and lobectomy. Finally, in 2022, the well-known Japanese trial and a German randomized study were published [8,9]. The Japanese Clinical Oncology Group and the West Japan Oncology Group published the results of the JCOG/WJOG 0802 trial that included 1106 patients with NSCLC up to 2 cm, located in the outer third of the lung without lymph node involvement. Overall survival at five years was the primary purpose, and the authors demonstrated equivalent survival between segmentectomy and lobectomy (94.3% and 91.1%, respectively). Despite this extraordinary result, some perplexities remain. Thus: (1) most deaths were unrelated to primary NSCLC; (2) Local relapses were significantly lower in the lobectomy arm (5% versus 11%); (3) Grade ≥ 2 air loss and the reinsertion of the chest tube rates were minor in the lobectomy arm (3.8% versus 6.5% and 1.4% versus 3.8%, respectively); and (4) predefined difference in median FEV1 reduction, which was set at 10% after a year, was not reached: the lobectomy arm experienced an advantage over the segmentectomy arm of a modest 3.5%, a clinically irrelevant difference.

The DRKS00004897 study was also published in 2022 [9]. This randomized trial, which was conducted in Germany, Austria, and Switzerland, was closed before reaching the predetermined sample size. The authors attribute the failure to reach the sample size to the strict inclusion criteria and the reluctance of surgeons to perform surgeries unsuitable for residents. However, we have included this study in our meta-analysis despite its failure to reach the planned sample size.

After our bibliographical research was completed, the results of the CALGB 140503 study were published [31]. This randomized trial recruited 697 patients from 83 institutions located in the United States, Canada, and Australia. Eligibility criteria included the presence of an NSCLC with a solid component up to 2 cm and located in the outer third of the lung; notably, the study also included wedge resections along with segmentectomies in the sublobar arm. Wedge resections were close to 60% of the study arm, and the authors did not perform a subgroup analysis to compare the two sublobar procedures. For this reason, the results of the CALGB study were not included in this meta-analysis.

For the first time, our meta-analysis adds two randomized trials to the observational studies that evaluated the oncological value of segmentectomy versus lobectomy in the treatment of NSCLC up to 2 cm. Despite a moderate heterogeneity, our hazard ratio analysis showed that overall survival was comparable between patients undergoing the two different surgical procedures. Looking at overall survival in terms of restricted mean survival time difference, the two procedures were still overlapping; importantly, though, despite the apparent overlap, the ratio of mortality risk was time-dependent. Figure 4 clearly shows that this ratio became significantly favored lobectomy starting 40 months after surgery.

Our DFS results were similar to those obtained for general survival; both HR and RMSTD analysis showed similar outcomes in patients who underwent lobectomy or segmentectomy. The only difference in overall survival could be detected by analyzing data by time: although the risk of recurrence increased over time for both procedures, the ratio between the two risks remained constant.

Segmentectomy is considered a more complex surgery and is burdened with a higher rate of postoperative complications than lobectomy. In our meta-analysis, the rate of complications tended to be higher in patients treated with segmentectomy, and, excluding Saji’s trial, statistical significance was reached. The increased rate of complications of patients undergoing segmentectomy was largely attributable to postoperative air leaks. Since the sectional technique of the inter-segmental plan has almost never been reported, we can only speculate that many of the authors have not used staplers. Following the inter-segmentation plan with a cutting tool is anatomically more correct than using a stapler but exposes the patient to possible prolonged air leaks.

Our meta-analysis has several limitations. First, the selected articles cover a rather wide time interval; during this interval, advances were made in the surgical, radiological, computer, and therapeutic fields. Although our statistical analysis does not show a change in the results over the years, we cannot exclude that technological advances may have introduced biases. We included observational studies that left the choice between the two procedures to the decision of the surgeons. The selection bias introduced by this recruitment method has been corrected by applying the propensity score matching analysis by some authors. Notwithstanding, this popular method of analysis has been criticized because it does not balance the entire vector of covariates and reduces the samples; accordingly, the results should be taken with caution [32]. Another limitation concerns the inclusion of observational studies based on large national registers. In these studies, reasons leading to the choice of surgical procedure are unclear, as is the possible conversion rate from segmentectomy to lobectomy [22]. Another limitation may be related to the definition and registration of postoperative complications. The mentioned limitations are intrinsic to observational studies and can only be corrected by the large number of patients included in our meta-analysis.

Despite these limitations, our meta-analysis strongly supports the idea that segmentectomy is oncologically equivalent to lobectomy in the treatment of NSCLC up to 2 cm. So, is this the time to change paradigms? Should the recently published randomized trial result lead us to change our attitude? We believe that many aspects still need to be clarified before unequivocally adopting segmentectomy as a substitute for lobectomy for small tumors located in the peripheral third of the lung. First, it is true that our meta-analysis of 5352 patients showed a similar risk of death between the two procedures, but looking at the risk ratio over time, it becomes evident that segmentectomy has a greater risk of death starting from forty months after surgery. Second, the location of the neoplasm in the peripheral third of the lung may not be considered homogeneous. Studies of lymphatic drainage of immediately subpleural lesions indicate that up to 66% of the drainage does not follow the bronchial route but runs subpleural, reaching the lobar hilum through pathways anatomically distant from the affected segment [33]. Third, our meta-analysis includes patients with variable percentages of solid and lepidic components; what proportion is best suited for a segmentectomy remains to be defined. Fourth, the specific characteristics of lung neoplasms, such as vascular, lymphatic, or air space invasion, are still not investigated in relation to the type of lung resection. Fifth, the molecular characteristics of the neoplasia will likely play a role in deciding which surgical procedures to prefer, especially in light of the likely introduction of induction therapy in the next future. Sixth and last, the debate remains open if the modest advantage in respiratory function of the segmentectomy justifies its application in relation to a survival deemed equal to that of the lobectomy.

In conclusion, adding recent randomized trials to selected observational studies, our meta-analysis confirms the oncological similarity of segmentectomy to lobectomy in terms of overall survival, at least for the first 40 months, and shows that disease-free survival is comparable in these two surgical procedures. We could not estimate the 30-day postoperative mortality, while the complications rate was higher, although not significantly, for segmentectomy. Our results are to be taken with caution, given the large number of items that still need to be addressed.

## Figures and Tables

**Figure 1 life-13-00947-f001:**
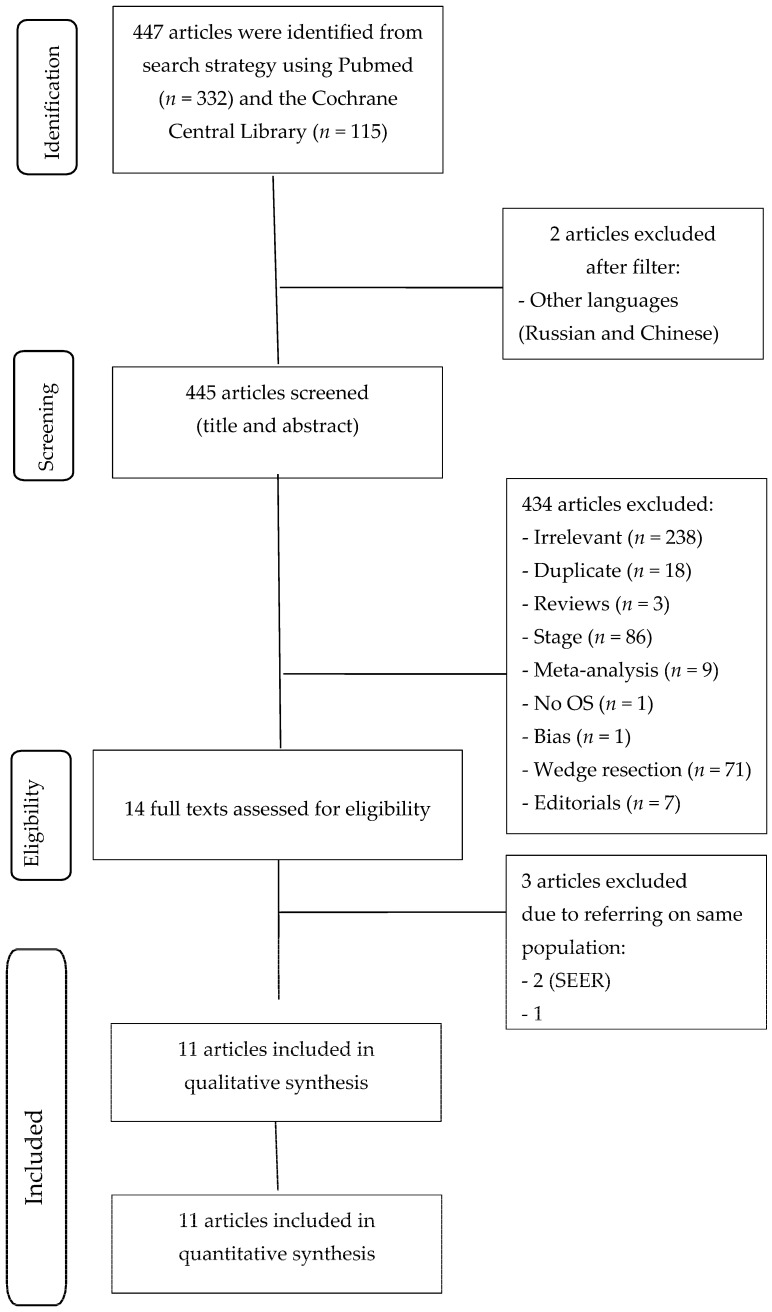
Prisma flow chart.

**Figure 2 life-13-00947-f002:**
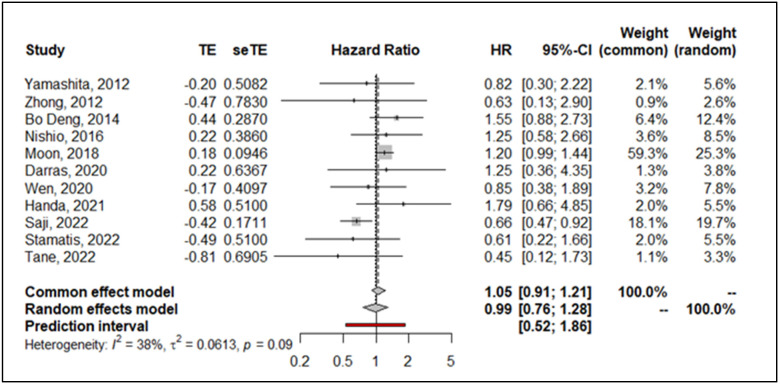
Forrest plot comparing segmentectomy to lobectomy for overall survival. (CI: confidence interval; HR: hazard ratio; se: standard error; TE: treatment effect) [8,9,18,19,20,21,22,23,24,25,26].

**Figure 3 life-13-00947-f003:**
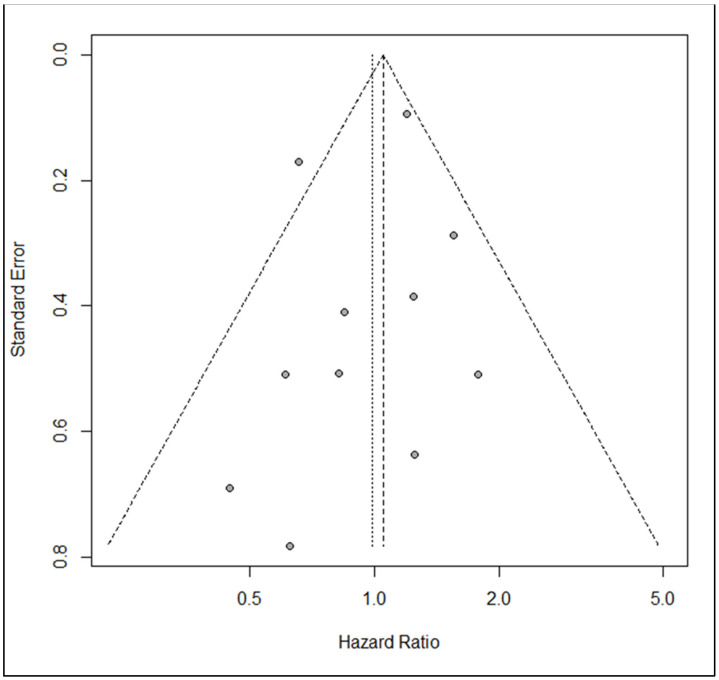
Funnel plot for overall survival.

**Figure 4 life-13-00947-f004:**
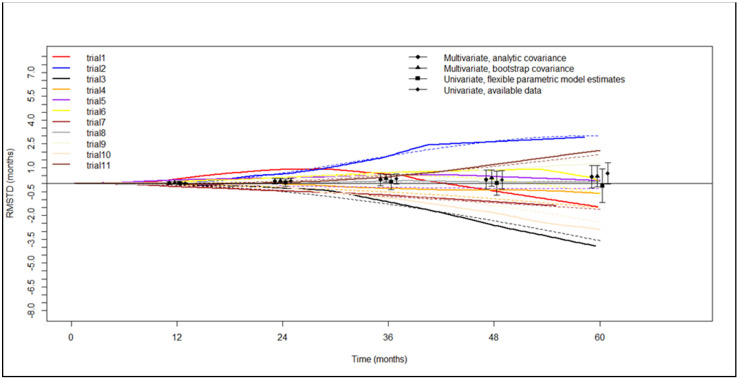
Graphical representation of restricted mean survival time difference with progressive time horizons for overall survival. RSMTD: restricted mean survival time difference. Trials are ordered as in the forest plot in Figure 2. Dots with vertical lines are punctual values with confidence intervals of the restricted mean survival time difference calculated at the indicated time points.

**Figure 5 life-13-00947-f005:**
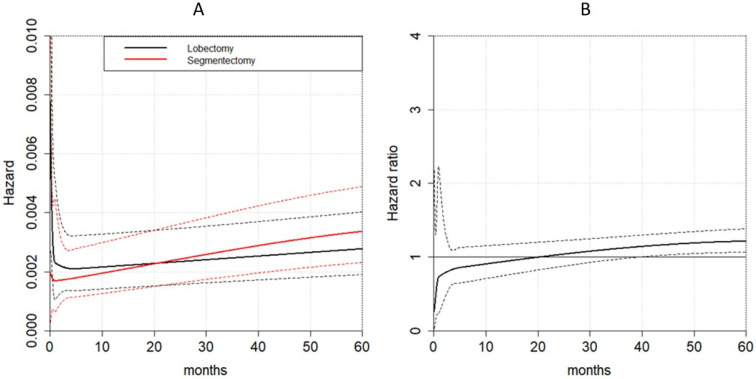
Time-dependent effects of the two surgical procedures on overall survival. (**A**): hazard for mortality for any cause; (**B**): hazard ratio for mortality between segmentectomy and lobectomy.

**Figure 6 life-13-00947-f006:**
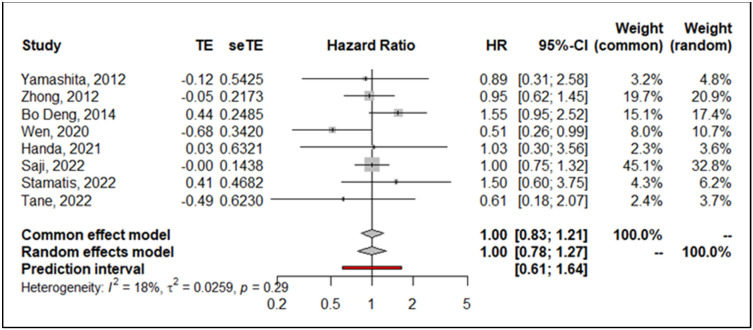
Forrest plot comparing disease-free survival in segmentectomy and lobectomy. CI: confidence interval; HR: hazard ratio; se: standard error; TE: treatment effect [8,9,18,19,20,23,25,26].

**Figure 7 life-13-00947-f007:**
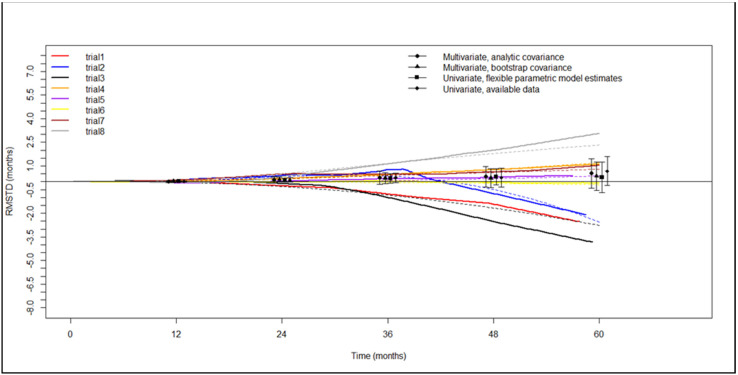
Graphical representation of restricted mean survival time difference with progressive time horizons for disease-free survival. RMSTD: restricted mean survival time difference. Trials are ordered as in the forest plot in Figure 5. Dots with vertical lines are punctual values with confidence intervals of the restricted mean survival time difference calculated at the indicated time points.

**Figure 8 life-13-00947-f008:**
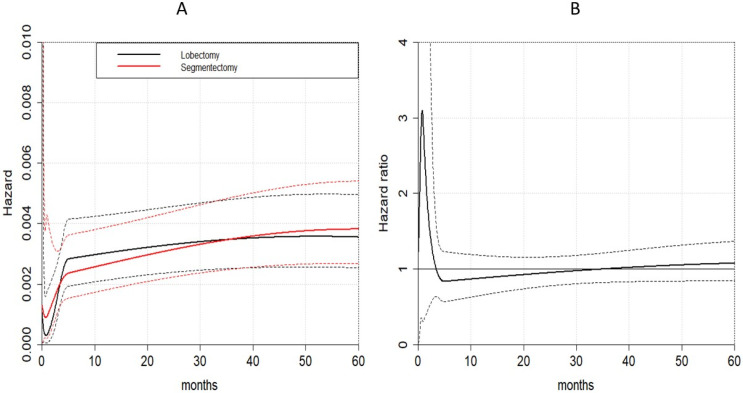
Time-dependent effects of the two surgical procedures on disease-free survival. (**A**): hazard for recurrence; (**B**): hazard ratio for recurrence between segmentectomy and lobectomy.

**Figure 9 life-13-00947-f009:**
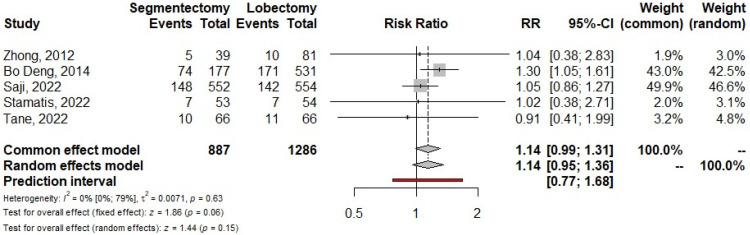
Forrest plot comparing segmentectomy to lobectomy for postoperative complications. CI: confidence interval, RR: relative risk [8,9,19,20,26].

**Table 1 life-13-00947-t001:** Characteristics of the included studies.

Authors, Year	Country	Years	Study Design	Patients, n		Female (%)		Mean Age (Range)		Smoking History	Comorbidities
				S	L	S	L	S	L		
Yamashita et al., 2012 [18]	Japan	2003–2011	R	76	72	N.A.	N.A.	N.A.	N.A.	N.A.	N.A.
Zhong et al., 2012 [19]	China	2006–2011	R	39	81	46	40.7	63.6 ± 8.0	64.9 ± 7.3	N.A.	Yes
Deng et al., 2014 [20]	USA	1997–2012	R, PSM	74	222	50	50.5	69.8 (11.9)	69.8 (10.1)	Yes	Yes
Nishio et al., 2016 [21]	Japan	1995–2009	P, PSM	59	59	35.6	35.6	64 (58–71)	61 (57–70.5)	N.A.	N.A.
Moon et al., 2017 [22]	Korea USA	2000–2014	R, PSM	809	809	65.6	65.8	67.8 ± 10.0	67.9 ± 9.5	N.A.	N.A.
Wen et al., 2020 [23]	China	2008–2018	R, PSM	214	214	67.3	66.8	59.3 ± 10	60.3 ± 10	Yes	N.A.
Darras et al., 2021 [24]	Switzerland	2014–2019	R	96	92	47	42	66.2 ± 10	63.6 ± 10.6	Yes	Yes
Handa et al., 2021 [25]	Japan	2010–2018	R	240	851	51.7	49.7	69 (62–74)	68 (61–73)	Yes	N.A.
Saji et al., 2022 [8]	Japan	2009–2014	RCT	552	554	47.5	47.1	67 (32–83)	67 (35–85)	Yes	Yes
Stamatis et al., 2022 [9]	Germany	2013–2021	RMT	53	54	39.6	44.4	69 (42–80)	66 (52–79)	Yes	N.A.
Tane et al., 2022 [26]	Japan	2007–2021	R, PSM	66	66	40.9	47–9	68.6 ± 8.5	68.1 ± 8.6	N.A.	N.A.

P: prospective trial; PSM: propensity score match; R: retrospective study; RCT: randomized controlled trial; N.A.: Not available.

**Table 2 life-13-00947-t002:** Newcastle–Ottawa Quality Assessment Scale.

Articles	Selection				Comparability		Outcomes			Total Quality Score
	Representa-Tiveness of Exposed Cohort	Selection of Non-Exposed Cohort	Ascertainment of Exposure	Demonstration That Outcome of Interest Was Not Present at Start of Study	Adjust for the Most Important Risk Factors	Adjust for Other Risk Factors	Assessment of Outcome	Follow-Up Length	Loss to Follow-Up Rate	
Yamashita et al., 2012 [18]	*	*	*	*	*	*	*	*	*	9
Zhong et al., 2012 [19]				*	*	*		*	*	5
Deng et al., 2014 [20]		*		*	*	*	*	*	*	7
Nishio et al., 2016 [21]	*	*	*	*	*	*		*	*	8
Moon et al., 2017 [22]	*	*	*	*	*	*		*	*	8
Wen et al., 2020 [23]				*	*	*		*	*	5
Darras et al., 2021 [24]	*	*	*	*	*	*	*	*	*	9
Handa et al., 2021 [25]	*	*	*	*	*	*		*	*	8
Saji et al., 2022 [8]	*	*	*	*	*	*		*	*	8
Stamatis et al., 2022 [9]	*	*	*	*	*	*	*	*	*	9
Tane et al., 2022 [26]				*	*	*	*	*	*	6

Quality of a study is judged on three broad perspectives. One * is one point, possible total points are 9.

**Table 3 life-13-00947-t003:** Restricted mean survival time difference for overall survival at different time horizons.

Time Horizon	No. Trials	RMSTD (Months)	SE	95% CI	*p* Value
12 months	11	0.03	0.04	From −0.04 to 0.11	0.369
24 months	11	0.14	0.09	From −0.04 to 0.32	0.123
36 months	11	0.25	0.18	From −0.12 to 0.62	0.185
48 months	11	0.26	0.30	From −0.34 to 0.87	0.397
60 months	7	0.44	0.35	From −0.26 to 1.13	0.218

RSMTD: restricted mean survival time difference; SE: standard error; CI: confidence interval.

**Table 4 life-13-00947-t004:** Restricted mean survival time difference for disease-free survival at different time horizons.

Time Horizon	No. Trials	RMSTD (Months)	SE	95% CI	*p* Value
12 months	8	0.00	0.02	−0.04 to 0.05	0.724
24 months	8	0.12	0.11	−0.09 to 0.33	0.275
36 months	8	0.25	0.21	−0.17 to 0.67	0.241
48 months	8	0.33	0.33	−0.32 to 0.97	0.324
60 months	4	0.54	0.48	−0.40 to 1.47	0.261

RSMTD: restricted mean survival time difference; SE: standard error; CI: confidence interval.

## Data Availability

Not applicable.

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
