# Peer review of "Oncological Outcomes of Segmentectomy versus Lobectomy in Clinical Stage I Non-Small Cell Lung Cancer up to Two Centimeters: Systematic Review and Meta-Analysis"

_life, 2023, doi:10.3390/life13040947_

Round 1

Reviewer 1 Report

This paper deals with a relevant topic. The large number of patients is clearly a major strength of the study.

The trouble with the studies comparing segmentectomy to lobectomy is that we tend to end up reproducing the same debatable results from which it remains difficult to draw a definite, all-encompassing conclusion. And so I think that what is important here beyond just crunching the numbers is how theoretical considerations factor into the design of the study and interpretation of the results.

When the authors mention (line 61) that segmentectomy « responds to the principles of an oncologically correct surgery »; what are these principles exactly, when referring to lung surgery, and why are they important? From a theoretical perspective, why would an anatomic resection be preferable to a non-anatomic (wedge) resection?

Sublobar resection is not a homogenous term. As the authors rightly point out later in the discussion, many studies include wedge resection. This is a major problem that has plagued the literature on the subject since the initial Lung Cancer Study Group publication. Likewise, « segmentectomy » is not a homogenous term either; what is more « limited »? A left upper division resection (S1,2,3), or a middle lobectomy, a common basal « segmentectomy » (4 segments) or a right upper lobectomy (3 segments)? I think these points would be interesting to point out in the introduction where the authors lay down the rationale for their study.

The authors’ literature search spans 3 decades. There is no question that segmentectomy has greatly evolved within that time. Current techniques using minimally invasive approaches and 3D imaging as well as localisation techniques are highly sophisticated and are difficult to directly compare with techniques used in the past. One should also recognize that the approaches for anatomic sublobar resection are not standardized, and that segmentectomy does require a (probably underreported) learning curve.

Although not immediately relevant to the study results, in my opinion any discussion of sublobar resection conjures up questions of evovling therapeutic alternatives, that are also gaining in popularity because of their favourable results and favourable risk benefit ratio, especially in this era of CT-screening.

I think that tumor size, although frequently used to justify sublobar resection, is a crude criterion. Clearly, lung cancer is a pathologically heterogenous disease (and in addition to classic morphologic distinctions, this increasingly includes immune-typing), and this will have to factor in some way into future decision-making algorithms involving lung cancer surgery. I think it is difficult to overstate the importance of pathology and also how information gleaned from preoperative imaging (radiomics) may contribute to operative decision-making.  

The one or two paragraphs in the discussion beginning aroung line 313 are important. I think the authors introduce extremely relevant topics that warrant further elaboration.

I am surprised that there were more complications associated with segmentectomy. This has not been the case in much of the published literature and has not been my experience. Perhaps this may be due to a higher proportion of air leaks (although this is highly technique dependent, and should not be the case when the intersegmental plane is stapled) and perhaps there is also a question of the learning curve.

In summary, this is an interesting study but I think that it should expand on some of the above themes in order to frame the results in a way that will make them more meaningful to the reader.

 NB: I did not find the figures anywhere

Author Response

Q: When the authors mention (line 61) that segmentectomy « responds to the principles of an oncologically correct surgery »; what are these principles exactly, when referring to lung surgery, and why are they important? From a theoretical perspective, why would an anatomic resection be preferable to a non-anatomic (wedge) resection?

A: We thank the Reviewer for this remark which is all but trivial.

The basic principles of oncological surgery go back many years and include the radical removal of the affected organ and its lymphatic drainage, adequate margins of resection, the removal of the disease en block and the attention to avoid the dissemination of neoplastic cells. Certainly, these principles have been declined differently according to the various organs and the various eras. Narrowing our attention to the lung, we can say that the segmentectomy seems to respond more correctly to the principles stated because, compared to the wedge resection, involves the removal of a well-defined anatomical unit that includes in addition to the dedicated vessels and bronchi also the respective lymphatic drainage.

We have implemented the following sentence:

Segmentectomy preserves healthy lung tissue protecting pulmonary function; at the same time, this procedure responds to the principles of an oncologically correct surgery such as the "en blocK" resection of the anatomical structure affected by the neoplasm and the respective lymphatic drainage. [Bremers AJ, Rutgers EJ, van de Velde CJ. Cancer surgery: the last 25 years. Cancer Treat Rev. 1999 Dec;25(6):333-53. doi: 10.1053/ctrv.1999.0147. PMID: 10644500]. Conversely, wedge resection leaves lymphatic drainage directed to the lung hilum in place, exposing the patient to local recurrence.”

Q: Sublobar resection is not a homogenous term. As the authors rightly point out later in the discussion, many studies include wedge resection. This is a major problem that has plagued the literature on the subject since the initial Lung Cancer Study Group publication. Likewise, « segmentectomy » is not a homogenous term either; what is more « limited »? A left upper division resection (S1,2,3), or a middle lobectomy, a common basal « segmentectomy » (4 segments) or a right upper lobectomy (3 segments)? I think these points would be interesting to point out in the introduction where the authors lay down the rationale for their study.

A: This remark is absolutely pertinent. We added a sentence as follows.

Moreover, segmentectomy cannot be considered equivalent for every portion of parenchyma and often the authors do not report which and how many segments have been removed (e.g., very common is the resection of the culmen, which actually includes three segments).”

Q: The authors’ literature search spans 3 decades. There is no question that segmentectomy has greatly evolved within that time. Current techniques using minimally invasive approaches and 3D imaging as well as localisation techniques are highly sophisticated and are difficult to directly compare with techniques used in the past. One should also recognize that the approaches for anatomic sublobar resection are not standardized, and that segmentectomy does require a (probably underreported) learning curve.

A: Undoubtedly the wide time span of bibliographic research includes advances in surgery, radiology, computer science, therapy, and understanding of lung cancer biology. Nevertheless, the statistical artifices we have implemented do not seem to show any change in clinical results over the years. However, we felt it was important to stress this point and added the following sentence in the limitations.

First, the selected articles cover a rather wide time interval; during this interval, advances were made in the surgical, radiological, computer and therapeutic fields. Although our statistical analysis does not show a change in the results over the years, we cannot exclude that technological advances may have introduced biases.”

Q: Although not immediately relevant to the study results, in my opinion any discussion of sublobar resection conjures up questions of evovling therapeutic alternatives, that are also gaining in popularity because of their favourable results and favourable risk benefit ratio, especially in this era of CT-screening.

A: This is a point we discuss every day during multidisciplinary meetings! More than one article would be needed to address this topic. We have merely introduced a hint.

In addition, alternative therapies such as radiosurgery or thermoablation seem to offer results at least comparable to surgery. To save functioning lung tissue,…”

Q: I think that tumor size, although frequently used to justify sublobar resection, is a crude criterion. Clearly, lung cancer is a pathologically heterogenous disease (and in addition to classic morphologic distinctions, this increasingly includes immune-typing), and this will have to factor in some way into future decision-making algorithms involving lung cancer surgery. I think it is difficult to overstate the importance of pathology and also how information gleaned from preoperative imaging (radiomics) may contribute to operative decision-making.

A: We are in complete agreement with the reviewer. Indeed, in the paragraph preceding the conclusions we introduced exactly these arguments. We fear that they cannot be further developed here, given the nature of the manuscript.

Q: The one or two paragraphs in the discussion beginning aroung line 313 are important. I think the authors introduce extremely relevant topics that warrant further elaboration.

A: We thank the Reviewer for his comment. Frankly, if the Reviewer and the Editor agree, we would not expand this section in order not to distort the intent of the manuscript. Expanding this section, which includes important discussion cues, without support from the selected articles, would turn the manuscript into an editorial. The points we list in these sentences are exactly what we expect to be clarified by future research. In any case, if deemed mandatory, we will argue each of the points listed in this section.

Q: I am surprised that there were more complications associated with segmentectomy. This has not been the case in much of the published literature and has not been my experience. Perhaps this may be due to a higher proportion of air leaks (although this is highly technique dependent, and should not be the case when the intersegmental plane is stapled) and perhaps there is also a question of the learning curve.

A: Once again we fully agree with the Reviewer. We added the following sentence.

The increased rate of complications of patients undergoing segmentectomy was largely attributable to postoperative air leaks. Since the sectional technique of the inter-segmental plan has almost never been reported, we can only speculate that many of the authors have not used staplers. Following the inter-segmentation plan with a cutting tool is anatomically more correct than using a stapler but exposes the patient to possible prolonged air leaks.”

Reviewer 2 Report

I could not find the figures (mentioned in the manuscript) an any of the attachment for review.

Ilaria Righi et al presented the result of a meta-analysis on the overall and disease-free survival after segmentectomy vs. lobectomy in Stage I NSCLC.

If you would like to list the 40-month HR as a significant finding (important enough to be highlighted in the abstract), then it should at least be part of the 2nd outcome in your methodology.

Author Response

Q: If you would like to list the 40-month HR as a significant finding (important enough to be highlighted in the abstract), then it should at least be part of the 2nd outcome in your methodology.

We thank the Reviewer for his/her suggestion.

We modified the following sentence.

The primary outcomes for pooled analysis were OS and disease-free survival (DFS), secondary outcomes were behavior of relative risk for OS and DFS over time, postoperative complications and 30-day mortality.”
